# Correlation of fitness landscapes from three orthologous TIM barrels originates from sequence and structure constraints

Yvonne H. Chan[1], Sergey V. Venev[2], Konstantin B. Zeldovich[2],** & C. Robert Matthews[1],**

Sequence divergence of orthologous proteins enables adaptation to environmental stresses and promotes evolution of novel functions. Limits on evolution imposed by constraints on sequence and structure were explored using a model TIM barrel protein, indole-3-glycerol phosphate synthase (IGPS). Fitness effects of point mutations in three phylogenetically divergent IGPS proteins during adaptation to temperature stress were probed by auxotrophic complementation of yeast with prokaryotic, thermophilic IGPS. Analysis of beneficial mutations pointed to an unexpected, long-range allosteric pathway towards the active site of the protein. Significant correlations between the fitness landscapes of distant orthologues implicate both sequence and structure as primary forces in defining the TIM barrel fitness landscape and suggest that fitness landscapes can be translocated in sequence space. Exploration of fitness landscapes in the context of a protein fold provides a strategy for elucidating the sequence-structure-fitness relationships in other common motifs.

[1] Department of Biochemistry and Molecular Pharmacology, University of Massachusetts Medical School, 364 Plantation Street, Worcester, Massachusetts 01605, USA. [2] Program in Bioinformatics and Integrative Biology, University of Massachusetts Medical School, 368 Plantation Street, Worcester, Massachusetts 01605, USA. ** These authors jointly supervised this work. Correspondence and requests for materials should be addressed to K.B.Z. (email: Konstantin.Zeldovich@umassmed.edu) or to C.R.M. (email: C.Robert.Matthews@umassmed.edu).

Proteins carry out diverse and essential functions in all living organisms. Notable among these functions is the catalysis of a host of complex chemical reactions under a broad range of environmental conditions. Based on enzyme classification, proteins catalyse over 5,700 unique biochemical reactions by employing ∼1,400 unique folds[1]. This structural redundancy shows that enzymes capitalize on robust structural platforms to introduce novel chemistries through sequence diversification. The TIM barrel fold is one of the oldest and most common motifs in biology[2]. The polarized structure has a well-ordered scaffold that is fused to an autonomous domain for substrate binding, catalysis, and product releases; the spatial segregation of the active site from the stabilizing protein core is an elegant solution for evolving new functions of TIM barrels[3]. The TIM barrel superfamily contains 57 distinct families and encompasses five of the six enzyme commission functional categories catalysing at least 34 unique functions[4].

As sequences and functions of TIM barrel fold proteins emerged through a divergent evolutionary process, a quantitative description of the TIM barrel fitness landscape would be a crucial step in our understanding of protein evolutionary dynamics. Recent deep mutational scanning experiments provided significant insight into the fitness landscapes of individual proteins[5–13] or very closely related homologues[14]. It was found that thermodynamic effects of mutations are not very sensitive to sequence background, prompting biophysical models of sequence evolution[10,15–19]. Furthermore, mutational scans showed that site-specific amino-acid preferences and, presumably, fitness landscapes are nearly identical in homologues of influenza virus nucleoprotein with 94% identity[14]. To date, it remains unclear whether the fitness landscapes remain similar in homologous proteins with a strongly divergent evolutionary history.

Here we perform a mutational scan of three orthologous TIM barrel fold proteins, indole-3-glycerol phosphate synthase (IGPS), to experimentally determine the fitness landscapes of proteins sharing the same fold and function, but with ancient divergences and low sequence identity. IGPS proteins were mutagenized in eight 10-residue segments, following the fold symmetry and covering the β-barrel core and adjacent elements of the αβ- and βα-loops. A tryptophan auxotrophic *S. cerevisiae* yeast strain, created by deletion of the endogenous IGPS gene, was transformed to prototrophy with each of the three orthologous genes. Relative fitness of the mutants, which depends on IGPS activity, was determined by the abundance of the mutant DNA sequence relative to wildtype (WT) over time[20].

A comprehensive analysis of 5,040 mutations in the three orthologues demonstrated that the fitness landscapes of IGPS orthologues are statistically significantly correlated to each other, despite the sequence identity of approximately 30–40%. Surprisingly, we found that fitness can be dramatically enhanced by mutations distal from the active site in all three orthologues. Fold geometry and structural elements of the TIM barrel fold, as well as sequence conservation and amino-acid biochemistry, all impose measurable constraints on the fitness landscape. Principal component analyses (PCA) detected commonality between sources of fitness variance in the three IGPS TIM barrel orthologues, while statistical coupling analysis (SCA) revealed evolutionary correlations between the active site and positions of the distal beneficial mutations. These results have significant implications for the design of TIM barrel enzymes with novel functions not found in nature and insights into the unanticipated allostery for the TIM barrel motif.

## Results

### Fitness landscape of individual IGPS TIM barrel proteins.
We applied the EMPIRIC deep mutational scan approach, developed

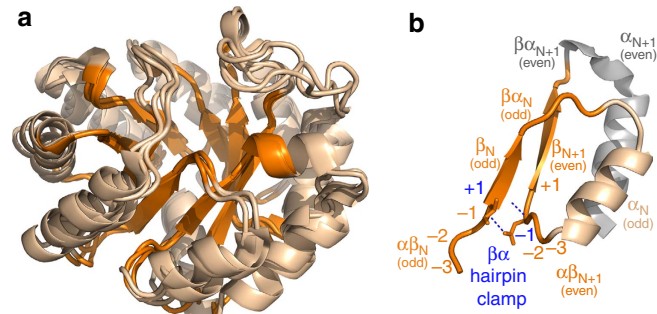

**Figure 1 | Orthologous IGPS proteins fold into highly similar canonical TIM barrel structures.** (**a**) Ribbon diagrams of structurally aligned SsIGPS (PDB: 2C3Z), TmIGPS (PDB: 1I4N), and TtIGPS (PDB: 1VC4). EMPIRIC mutagenesis library positions are highlighted in orange. The parallel β-strands vary in length from 4–6 residues and the β-barrel structure forms four layers of side chains from alternating β-strands that protrude into the protein core (Supplementary Fig. 1). Preceding the N terminus of the β-strands, short αβ-loops, generally 3–4 residues in length, are proposed to play a role in stability[74]. At the C-terminus of the β-strands, long-βα loops, between 5 and 13 residues, link the C termini of the β-strands to the N termini of the subsequent α-helices and invariably form the active site of the enzyme. (**b**) The four-fold symmetric βαβα repeat unit holds the minimal independently folding unit, the βαβ module, highlighted in orange, wheat, and orange, respectively. The even helix shaded in grey does not contribute significantly to stability of the four-fold unit. Within the βαβ module, a canonical βα hairpin clamp formed between a main chain amide H-bond donor in +1 position of the odd β-strand and the side chain acceptor at the −1 position in the following even αβ-loop is highlighted in blue.

by the Bolon group[20], to explore the fitness landscape of a monofunctional enzyme of the TIM barrel fold. In the canonical TIM barrel structure, eight β-strands and α-helices alternate in sequence, and the β-strands assemble sequentially into a cylindrical core around which the α-helices form a helical shell (Fig. 1a). The TIM barrel scaffold is highly symmetrical, displaying a four-fold βαβα symmetry at the level of the smallest independent folding unit, the βαβ module (Fig. 1b)[21]. We chose three phylogenetically divergent IGPS orthologues from a thermophilic archaeon, *S. solfataricus* (SsIGPS), and two thermophilic bacteria, *T. maritima* (TmIGPS) and *T. thermophilus* (TtIGPS), Table 1. For the three proteins, we carried out selection experiments in yeast to determine the fitness of all possible mutations in libraries of 10 positions spanning the 8 αβ-loops, β-strands, and initial portion of the βα-loops (Fig. 2). Mutations were introduced by restriction of the gene followed by its ligation with a cassette containing all 64 codons for a given position, as opposed to transcribing the gene using an error-prone polymerase[9,20]. This approach generates a well-defined, rather than random, sequence diversity in the mutagenized region, reducing experimental noise.

Fitness was quantified as the selection coefficient *s*, that is, the slope of the relative abundance of mutant to WT IGPS over time (see Methods and Supplementary Fig. 2). Mutants with selection coefficient of 0 have doubling time equal to WT, while those with a selection coefficient of −1 are lethal. Over 55% of the mutations were deleterious with $s < -0.75$. The β-strands showed a particularly low tolerance to mutation. Presumably, mutations in the core destabilize the native state, resulting in a lower population of functional IGPS enzymes, or distort the active site, reducing catalytic power. Six catalytically important residues form the active site and are located at or near the C termini of β1, β3, β5, β6 and β7. Mutations at these highly conserved residues

were strictly not tolerated. Bimodal distributions are often observed with fitness landscapes, describing a thermodynamic 'cliff' at which proteins unfold and the mutations are lethal[17,19,22]. While we observed the expected bimodal distribution, the high

fitness mode showed a bias towards beneficial rather than neutral fitness (Supplementary Fig. 3). Positive selection coefficients were associated with several mutations in the active site βα-loops at positions that do not directly support enzyme chemistry. Remarkably, fitness can be increased by mutations far from the active site in the αβ-loops, most notably at the positions of βα hairpin clamps (Fig. 2). βα hairpin clamps are long-range hydrogen bonds between the even numbered αβ-loops and the preceding odd numbered β-strands, enforcing β-strand alignment and providing stability (Fig. 1b)[23]. Based on the repeating fitness patterns in the minimally independent folding βαβ modules contained within the four-fold βαβα symmetrical units for each of the three orthologues, we analysed the selection coefficients in groups of two consecutive libraries, that is, odd (n) and even (n + 1) strands (Fig. 1b).

**αβ-loops.** Within the canonical TIM barrel structure, there is a bias towards the glycine-x-aspartic acid motif in the αβ-loops leading up to the even number β-strands[2,23]. In the even numbered αβ-loops, mutations away from small hydrophobic residues at −2 position (X in the motif) with respect to the β-strand, prevent the tight turn required to dock the preceding helix. We found that this distortion of the βαβα module results in low organismal fitness, likely reflecting destabilization of the native state from poor packing. In contrast, the odd numbered αβ-loops link adjacent modules and are less constrained in their residue choice. Many mutations at the −3 and −2 positions of the odd strands were beneficial. At the −1 position, the higher fitness in the even versus the odd strand reflects their different accessible surface area (ASA). The average selection coefficient of these positions is correlated with the ASA, where mutations at the more buried even loops resulted in lower fitness than mutations in the more solvent exposed odd loops (Supplementary Fig. 4). Similar results were found using the relative surface area (RSA)[24] instead of ASA.

**Table 1 | Pairwise sequence and structure similarity of the three IGPS orthologues and mutagenized libraries.**

|  | SsIGPS | TmIGPS | TtIGPS |
|---|---|---|---|
| SsIGPS | — | Library sequence: Identical: 51% Similar: 71% | Library sequence: Identical: 49% Similar: 68% |
| TmIGPS | Full-length sequence: Alignment length: 267 Identical: 30% Similar: 49% Structure: Alignment length: 216 r.m.s.d.: 1.58 Å | — | Library sequence: Identical: 54% Similar: 73% |
| TtIGPS | Full-length sequence: Alignment length: 271 Identical: 35% Similar: 49% Structure: Alignment length: 214 r.m.s.d.: 1.24 Å | Full-length sequence: Alignment length: 277 Identical: 27% Similar: 43% Structure: Alignment length: 241 r.m.s.d.: 1.72 Å | — |

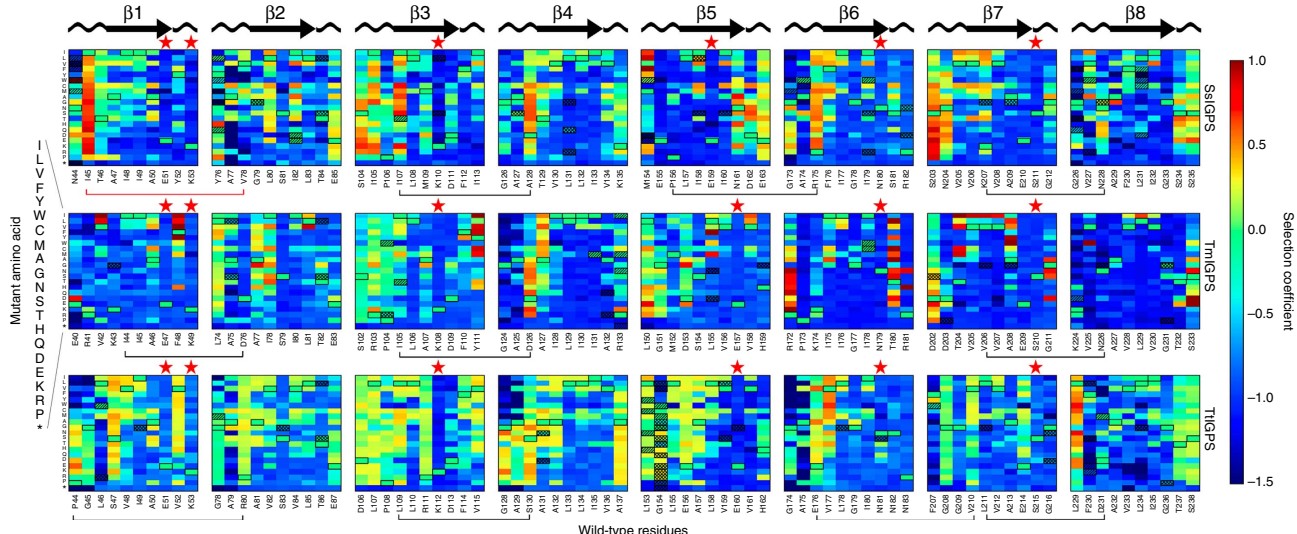

**Figure 2 | Fitness landscapes of the three IGPS orthologues.** Values of the selection coefficient are colour-coded on a continuous scale from 1 to −1.5 indicated by the colorbar. WT residues and positions are labelled at the bottom of each panel. Mutant amino acids are indicated on the vertical axis. Within the heatmap, WT residues are indicated by the black outline. Low-quality data filtered from analysis are indicated by the red checkered boxes. Canonical secondary structures are drawn at the top of the panels. Active sites are indicated by the red stars at the top of the position columns. βα hairpin clamps are indicated by the black brackets at the bottom of the position columns. Fitness gains were observed with several mutations of the βα hairpin clamps. For example, more than half the mutations in the three β3α3 hairpin clamps SsIGPS I107 and D128, TmIGPS I105 and D126, TtIGPS I109 and S130 are beneficial (s > 0). Two non-canonical interactions analogous to the βα hairpin clamp in SsIGPS are indicated by the red brackets. An ionic interaction is observed between E155 and R175. A hydrophobic stacking interaction is observed between I45, V78, and F40 (not included in library). All mutations of SsIGPS I45, except to stop codons, resulted in fitness advantage over WT.

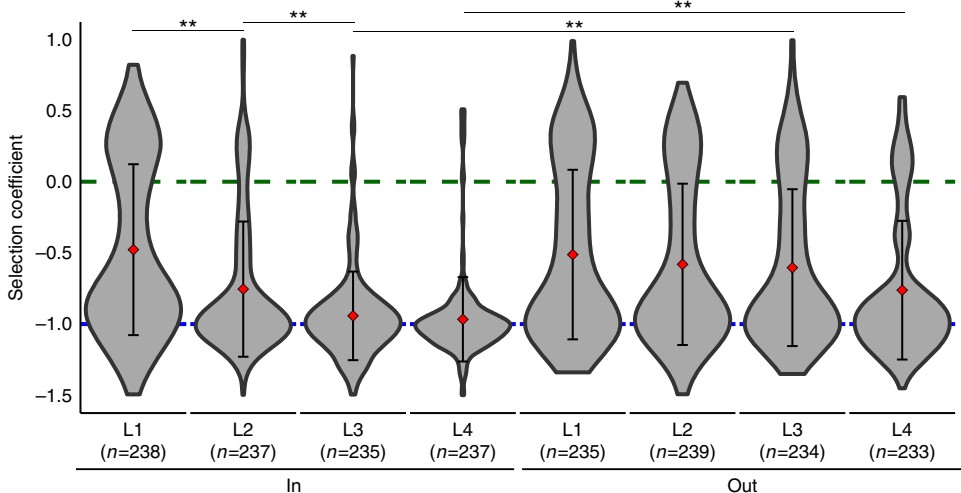

**Figure 3 | Distribution of fitness values displayed as violin plots stratified by side chain orientation 'in' (left) and 'out' (right) and by layers (L1 to L4).** Guide lines are drawn at $s = 0$ in green and $s = -1$ in blue. The mean and s.d. are indicated by the red diamond and error bars. Sample numbers are indicated beneath each layer. Permutation tests ($n = 10,000$) were used to assess statistical significant differences between distributions based on ratio of beneficial ($s > 0$) to total mutations (**$P < 0.02$). See text and Supplementary Fig. 1 for details.

**βα hairpin clamp**. Loss of the βα hairpin clamp between the canonical aspartic acid at the −1 position of the even strand and the main chain amide hydrogen beneath a large hydrophobic side chain at the N terminus of the preceding odd numbered β-strand elicited a fitness gain relative to the WT protein (Figs 1b,2). This enhancement is the most dramatic at the β3α3 hairpin clamp in all three orthologues. Significantly, mutation of I107 in the SsIGPS β3α3 hairpin clamp results in high fitness, while just one position away in the β-barrel, L108, mutations are mostly deleterious. The distinctly different responses highlight the different roles of the αβ-loop and the β-strand in protein stability and enzymatic function, each driving fitness of the host organism (Fig. 2). The most beneficial mutation in SsIGPS was I45Q, with selection coefficient of $+0.89$. Position I45 is involved in a triple stack of hydrophobic side chains where F40 (α0) is sandwiched between I45 (αβ0-loop) and V78 (αβ1-loop). This bridge between strands β1 and β2 is an alternative solution to the βα hairpin clamp typically found to stabilize odd and even β-strands. Mutations of the I45 to any other amino acid, except stop codons, are uniformly beneficial (Supplementary Fig. 5).

**β-strands**. The largely deleterious mutations in the β-strands indicate that minimal changes are tolerated within the protein core. Similar to the αβ-loops, the residues in each β-strand can be further stratified by side chain orientation into or out of the β-barrel and by participation in the four layer levels within the β-barrel (Supplementary Fig. 1). The evolved structure maximizes buried surface area of side chains, while minimizing steric clashes, by alternating side chain orientation between odd and even strands within a layer and between layers within β-strands[25,26]. Side chains that point inward form a highly stable hydrophobic core, while side chains pointing outward provide docking surfaces for the concentric α-helices by hydrophobic interactions. For side chains pointing outward in layers 2–4, a significant fraction of mutations is beneficial, in contrast to the almost complete lethality of their inward facing counterparts (Fig. 3). Mutations of β-strand residues pointing out of the barrel would be expected to perturb the α-helix and β-strand interface as well as the intervening βα-loop. Interestingly, inward facing side chains in layer 1 can also increase the fitness for a significant fraction of the mutations. For mutations in the fourth

layer, the poor fitness may reflect the known sensitivity of the βα-loops to the interface between the stability core and the active site loops[27].

**βα-loops and the active site**. The active site spans multiple β-strands and subsequent βα-loops (Fig. 2). Although mutations at catalytically important residues were uniformly deleterious, some nearby mutations showed improved fitness over WT. Across the three orthologues, beneficial mutations were observed at position 52 (SsIGPS numbering) between the glutamic acid (E51) and lysine (K53) active site residues bridging the β1-strand and βα1-loop interface. Coordination of the substrate to its reactive conformation is mediated by electrostatic interactions between K53 in the βα1-loop and K110 in the βα3-loop[28]. However, molecular dynamics simulations of thermophilic SsIGPS under mesophilic temperatures suggest that strong electrostatic interactions between K53 and the substrate result in a nonproductive substrate geometry[29]. Our fitness results suggest that disrupting the ionic interactions between E51 and the two lysines, K53 and K110, may favour the reactive conformation of the substrate. Other beneficial mutations include the phosphate binding residue, S234, and its adjacent residue, S235. These residues, located in the eighth 'βα-loop' form a 3[10]-helix, and mutation of either may distort the phosphate binding pocket so as to improve substrate binding and/or product release.

**Correlation of fitness landscapes between IGPS orthologues.** A major question emerging from this screen is whether protein fitness landscapes are conserved across sequence and/or structure space. We compared the fitness landscapes of the orthologues by calculating the Pearson correlation coefficient $R$ between fitness values of the 20 mutant amino acids at a pair of positions in two orthologues. The probability distribution of $R$ for a specific set of positions was then used to assess the similarity of fitness landscapes. We considered the following four sets of positions: (a) identical WT amino acids in a pair of orthologues irrespective of structural alignment; (b) all structurally aligned positions irrespective of WT amino acid; (c) all structurally aligned position with non-identical WT amino acids; and (d) positions aligned by their four-fold symmetry in the TIM barrel, irrespective of WT amino acid, Fig. 4.

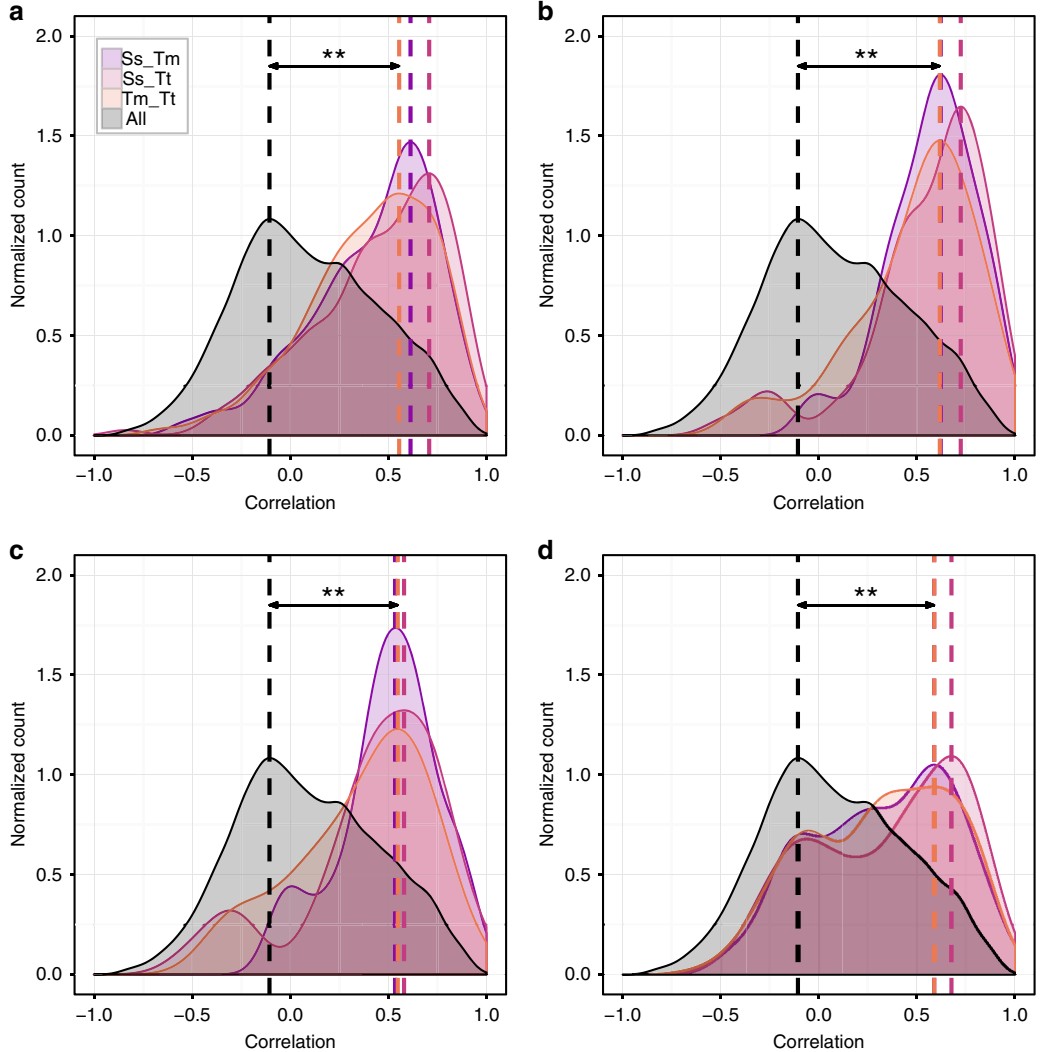

**Figure 4 | Fitness landscapes of orthologous proteins are correlated despite their low sequence identity.** Distributions of Pearson correlation coefficients between fitness landscapes of (**a**) identical WT amino acids in a pair of orthologues irrespective of structural alignment, (**b**) all structurally aligned positions irrespective of WT amino acid; (**c**) all structurally aligned position with non-identical WT amino acids, and (**d**) positions aligned by their four-fold symmetry in the TIM barrel, irrespective of WT amino acid. Fitness landscapes are statistically significantly correlated despite the low sequence identity and ancient divergence of IGPS orthologues (**$P<0.02$). Statistical significance is detailed in Supplementary Table 1.

In all cases, the fitness landscapes of these groups were significantly different from the null model (pairwise correlations between all positions) according to Kolmogorov–Smirnov test ($P<10^{-4}$, Supplementary Table 1). The correlations in all three pairs of orthologues were similar in magnitude, and in most cases their distributions were statistically indistinguishable (KS test, $P>0.05$, Supplementary Table 2). The modes of these specific distributions corresponded to the Pearson correlation between $\sim 0.5$ and 0.7. On the other hand, distributions of correlations for different sets of positions (for example, (a) versus (b) for all orthologues) were different in most cases (KS test, $P<10^{-4}$, Supplementary Table 3), suggesting that structure and sequence features define the fitness landscape. These results are statistically significant compared to variation between experimental replicates (see Methods and Supplementary Figs 9 and 10).

Despite ancient divergence and bacterial versus archaeal origins, we found that fitness landscapes of three IGPS proteins are significantly correlated. Remarkably, for positions exhibiting as low as 40% sequence conservation in the multiple sequence alignment of IGPS proteins, experimentally determined fitness landscapes of the three orthologues remained correlated to each other, $R \geq 0.4$

(Fig. 5a), complementing earlier observations using sequence alignments and mutational scans[14,30].

**Correlation of fitness landscapes and epistasis.** The correlations between fitness landscapes of orthologous proteins are intricately connected to the epistatic interactions in these proteins. If epistasis was absent, we would expect a near-perfect correlation between landscapes, as corresponding sites would have the same amino-acid preferences. Conversely, if epistasis was very strong, the fitness effects of every mutation would be entirely determined by the protein as a whole, and one would expect the fitness landscapes to become uncorrelated. Loosely, correlation of fitness landscapes is inversely proportional to the degree of epistasis. To unravel potential epistatic interactions in IGPS, for each orthologue, we selected 'transformative' mutations that correspond to WT residues in another orthologue, 'transforming' one protein into another. The distribution of selection coefficients of transformative mutations compared to the distributions of all selection coefficients in the three orthologues is significantly different (Fig. 5b). Transformative mutations are significantly enriched in

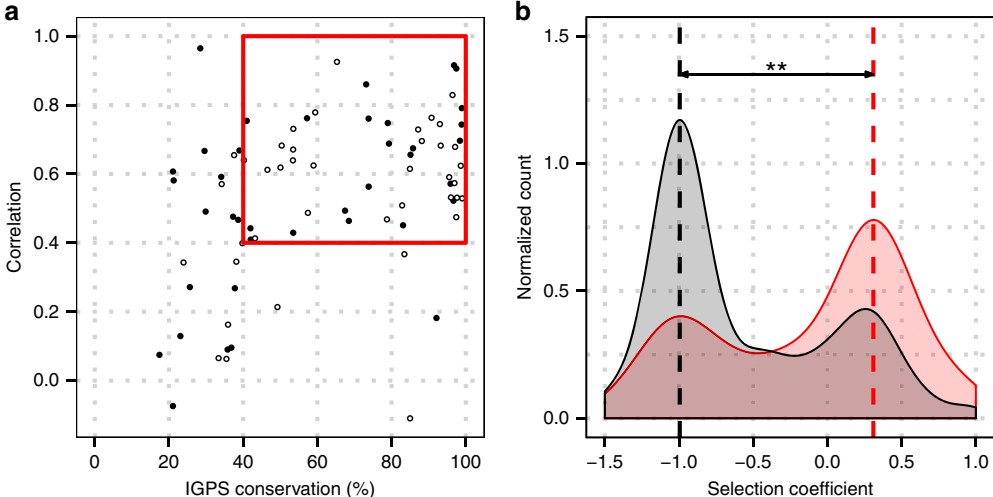

**Figure 5 | Sequence conservation and epistasis affect fitness landscapes.** (**a**) Fitness landscapes of IGPS orthologues are significantly correlated ($R > 0.4$) for IGPS positions ('open circle' odd numbered libraries, 'filled circle' even numbered libraries) displaying 40 to 100% conservation in the multiple sequence alignment (red box): the fitness landscapes of highly conserved positions are strongly correlated. At the same time, the wide range of correlations before the 'cliff' of 40% conservation suggests that sequence conservation is not the only determinant of fitness. (**b**) Mutations that 'transform' one IGPS orthologue into another primarily have neutral or beneficial effects (pink histogram) compared to all mutations (grey). Epistatic interactions are responsible for the existence of transformative detrimental mutations (minor peak on the pink histogram around $s = -1$). The distribution of selection coefficients for the transformative mutations is significantly different that than of the null distribution (**$P < 0.02$).

neutral and beneficial mutations; deleterious mutations are depleted. As widely expected, an amino acid that has naturally evolved in one of the orthologues is often tolerated in another orthologue. However, the existence of detrimental transformative mutations suggests epistatic interactions, as their fitness is strongly affected by interactions with the rest of the protein. We found that about 31% of the transformative mutations were detrimental ($s < -0.5$) in IGPS. For comparison, a previous study with isopropymalate dehydrogenase from *E. coli* and *P. aeruginosa* found that 38% of 168 transformative mutations were detrimental *in vitro*[31].

**Sources of variance in experimental fitness landscapes of IGPS.** To determine if the major sources of fitness variance are equivalent across the three orthologues, we applied PCA to the $80 \times 20$ matrices of selection coefficients of all mutations at each site surveyed, Supplementary Fig. 6. For each orthologue, the first two principal components accounted for almost 70% of observed fitness effects. The first principal component (PC1) explains ~51% of the fitness variance and, by construction, is proportional to the average fitness of the position (Fig. 6a). We did not detect a significant linear relationship of PCA components to features such as ASA, RSA, b-factor, and WT amino-acid sequence conservation, if all three orthologues are considered together (Fig. 6b). Still, a periodicity observed in the PC1 scores along the library positions prompted a closer look at the relationship between PC1 and the four-fold symmetric structure (Fig. 6c). This scatterplot of PC1 showed differences in the deviation of the principal component scores along the coarse-grained ten library positions. The highest scores, representing the highest average fitness, are found mainly at the first three positions of the odd strand libraries and the last position of the even strand libraries, associated with the αβ-loops and βα-loops, respectively. The lowest scores, representing the lowest average fitness, were found at positions six and eight, associated with the β-strands whose residues point into the β-barrel. The average four-fold PC1 varied linearly with average four-fold IGPS conservation ($R = 0.84$) (Fig. 6d). Thus, fitness must relate to

structural conservation of the four-fold symmetry. The second principal component (PC2) explains ~17% of the variance in our fitness data set and is correlated best with the hydrophobicity of the WT residues ($R_{\text{EMPIRIC PC2 to Hydrophobicity}} \sim 0.67$). Hydrophobicity of the WT residues is a major factor affecting the fitness outcome, independent of the residue position in the structure.

**Experimental data versus evolved IGPS and TIM sequences.** To separate the factors that are important for function from those important for stability in our fitness assay, we performed separate PCA analyses of representative sequences of IGPS and TIM barrel proteins, using amino-acid frequencies at each site as a proxy for their evolutionary fitness (see Methods). Since all IGPS enzymes are TIM barrel proteins, we expected that the major drivers of sequence variance will reflect stability and structure for the TIM PCA and stability, structure, and function for the IGPS PCA. By selecting the 80 aligned positions examined in our fitness assay, we directly compare and contrast the experimental EMPIRIC IGPS fitness landscape to the naturally evolved amino-acid preferences in IGPS and TIM barrels.

The first principal component for both the IGPS PCA and TIM PCAs accounted for ~30% of the variance found in their amino-acid preferences and was correlated with WT amino-acid hydrophobicity ($R_{\text{IGPS PC1 to hydrophobicity}} \sim 0.70$). The two components correlated with each other ($R_{\text{IGPS PC1 to TIM PC1}} \sim 0.71$) and with the second component of the EMPIRIC PCA ($R_{\text{EMPIRIC PC2 to IGPS PC1}} \sim 0.69$, $R_{\text{EMPIRIC PC2 to TIM PC1}} \sim 0.59$), which correlated with hydrophobicity. Thus, stability appears to be the main driver for the residue selection within IGPS and TIM barrels, in general, for the segments examined.

The second principal component accounts for ~20% of the variance observed in IGPS and TIM barrels and is collinear with information content of the sequence alignment, indicating conservation. Similar to PC1, the two PC2 of the IGPS PCA and TIM PCA were correlated ($R_{\text{IGPS PC2 to TIM PC2}} \sim 0.68$). At the same time, no correlation was observed between TIM conservation and EMPIRIC fitness in the IGPS orthologues. Presumably,

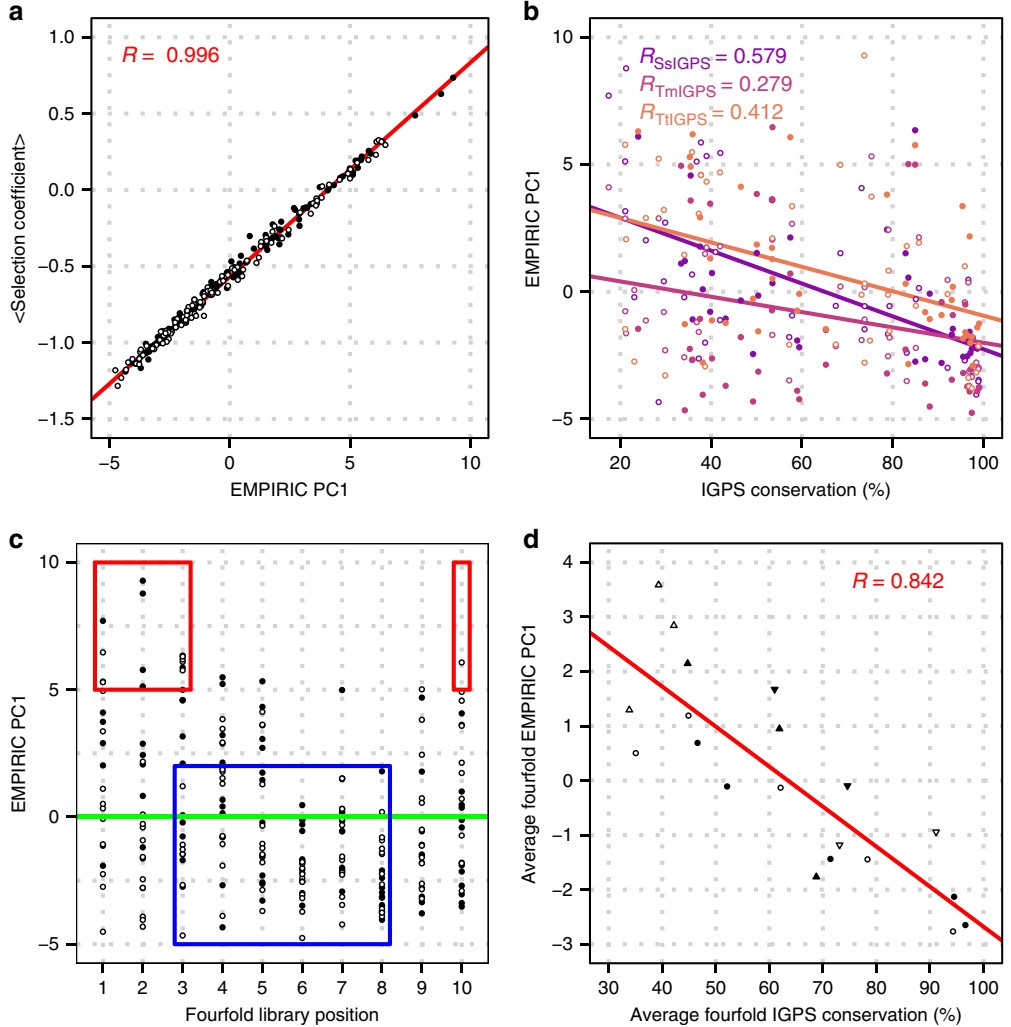

**Figure 6 | The first principal component of fitness landscape is related to average four-fold conservation.** (**a**) By construction, the first principal component (PC1) of the EMPIRIC PCA is linearly related to average fitness ($R = 0.996$, 'open circle' odd numbered libraries, 'filled circle' even numbered libraries). (**b**) A linear relationship between EMPIRIC PC1 and IGPS conservation of varying strength was observed for each orthologue. A linear correlation of $R = 0.408$ was observed if all three orthologues were considered together. (**c**) The values of EMPIRIC PC1 vary with four-fold aligned positions, implicating structure in fitness determination. The canonical secondary structures are indicated above the plot. The green line indicates the average score. Highest scores were observed at both ends of the library positions, associated with the αβ and βα-loops (red boxes). Lowest average scores were observed in the intermediate positions, associated with the β-strands (blue boxes). (**d**) Average EMPIRIC PC1 scores based on the four-fold alignment correlate linearly with average four-fold IGPS conservation ($R = 0.842$).

conservation in a sample of 71 representative TIM sequences only reflects very general fold patterns, and has little predictive value for the fitness of mutants in the specific family such as IGPS.

**Statistical coupling analysis.** While active site residues are highly conserved to preserve enzymatic function, other sites indirectly supporting enzyme activity or those supporting protein stability can also show position-specific sequence constraints and correlated substitutions between positions within a protein family[32–34]. SCA was used to characterize functionally important co-evolving residues in SsIGPS that may inform our fitness results[32]. Two significant sectors were identified. Sector one (46 positions) involved mainly amino acids at the interface between the β-barrel and α-helical shell (Supplementary Fig. 7a). The distribution of fitness effects for the 17 positions with known fitness was unimodally deleterious. Over 37% of the residues were branched aliphatic residues and 28% were charged residues, highlighting

the importance of hydrophobic and electrostatic interactions in stabilizing the β-barrel/helical shell interface throughout evolution. In contrast, sector two (45 positions) described both stability and functional properties. The distribution of fitness effects for the 25 positions with known selection coefficients was bimodal, $s_{\mathrm{modes}} \sim -0.9$ and $-0.3$, reflecting highly conserved sites that are intolerant of mutations and sites of high adaptation potential. Active site residues, including the substrate-binding site, made up $\sim 20\%$ of the sector (Supplementary Fig. 7b). Of particular interest, two non-active site positions found to improve SsIGPS fitness with mutation were identified in sector two, residues I45 and D128. Corresponding mutations of the β3α3 hairpin clamp in the other two orthologues yielded similar fitness gains. Serving as a putative conduit between the active site and these two residues are select α-helical and β-strand residues in the SCA that span the two ends of the protein (Supplementary Fig. 8). The distributed nature of covariation, or SCA sectors, over the IGPS structure is consistent with statistical observations of long-range effects of mutations[35].

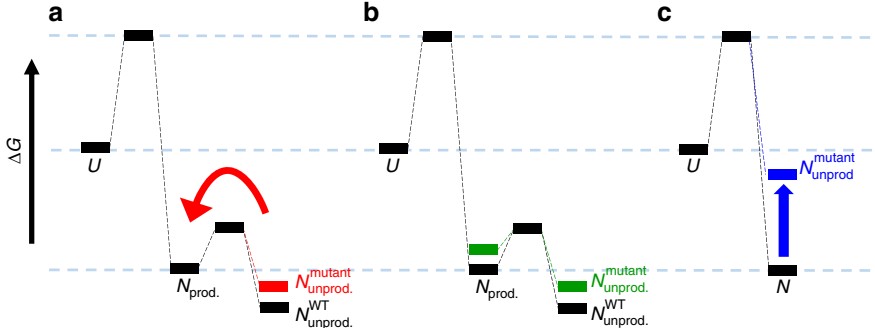

**Figure 7 | Putative effects of mutations on the energy landscape of IGPS.** Free energy diagrams showing three possible scenarios for the effect of a single mutation on the folding free energy surface ($\Delta G°$, Gibbs free energy of folding; $U$, unfolded state; $N$, native state ensemble). An ensemble of rapidly interconverting productive ($N_{prod.}$) and nonproductive ($N_{unprod.}$) conformations resides in the native basin. At mesophilic temperatures, thermophilic IGPS access the productive conformation to a lesser extent than at their native thermophilic temperatures. (**a**) Beneficial mutations may destabilize the unproductive native state without a concomitant destabilization of the higher energy, productive conformation, resulting in a shift in the population from the unproductive to the productive conformation. Increased activity improves fitness. (**b**) WT-like mutations may destabilize both the unproductive and productive states, resulting in no net change in the population. No change in fitness is observed. (**c**) Deleterious mutations may greatly destabilize the native state, resulting in a population shift to inactive, partially or fully unfolded, states that are susceptible to proteolysis. Poor fitness is associated with loss of enzymatic capabilities.

## Discussion

Systematic exploration of fitness landscapes has become possible using EMPIRIC and other high throughout mutagenesis approaches[9–14,20,36]. We have explored the fitness landscapes of three orthologous TIM barrel proteins from archaea and bacteria to answer two important questions at the intersection of biophysics and evolutionary biology: (1) what are the salient sequence and structural correlates of a TIM barrel fitness landscape, and (2) are the fitness landscapes of phylogenetically divergent orthologous proteins correlated with each other and, if so, what is the basis of the correlation?

Protein fitness landscapes depend on stability and enzymatic function, which are both affected by sequence and structure[3,13,18]. Thermophilic proteins employ multiple mechanisms to maintain their structure at high temperatures, including a higher content of salt bridges and H-bonds relative to their mesophilic counterparts[37,38]. Increased stability is certainly required at high temperature, but it can be a liability at lower temperatures[39]. The native basin of proteins and enzymes contains a dynamic ensemble of rapidly interconverting states, some of which may be more favourable for particular functions, for example, substrate binding or the catalysis of chemical reactions[40]. For IGPS, ring closure of the substrate, 1-(o-carboxyphenylamino)-1-deoxyribulose 5-phosphate (CdRP), requires a minimal distance and favourable orientation between C1 and C2' for the reaction to proceed[28,29]. This conformation is mediated by electrostatic interactions between the substrate and the active site lysine residues[29]. At lower temperatures, the conserved lysine residues show decreased flexibility and restricted structural orientation leading to a greater population of CdRP in an extended, unproductive conformation. At higher temperatures, electrostatic interactions between IGPS and CdRP favour a reactive substrate conformation[29]. Access to these productive higher energy states in the native basin would increase if a mutation leads to a destabilized unproductive native state conformation without a concomitant destabilization of the productive higher energy state (Fig. 7a,b). We hypothesize that fitness gains are driven by increased local flexibility and/or dynamics accompanied by a population shift toward the productive catalytic conformation(s). Conversely, deleterious effects may be caused by a decrease in stability and the concomitant decrease in the concentration of the enzyme or by distortion of the active site and ensuing reduction of catalytic properties (Fig. 7c). In fact, largely aliphatic residues in the barrel interior (Supplementary Table 4) are essential for structural integrity and incur a high proportion of deleterious mutations (Fig. 3). Native state hydrogen exchange protection patterns in the HisF TIM barrel from *T. maritima* revealed that layers 2 and 3 were strongly protected from solvent exchange, implying a major role in stabilizing the native state[41]. The side chains in these two layers participate in large clusters of densely packed ILV residues, known to form cores of stability in globular proteins[42]. We argue that the core of the barrel is critical for the stability of both the unproductive and productive conformations of TIM barrels.

Increased activity of mutant thermophilic proteins, including IGPS, at a lower temperature has been previously reported, and typically involves modifications to the active site or substrate-binding site[43]. In contrast, many positions where mutations increased the fitness in our assay were distal to the active site in the βα-loops. For example, deletion of the canonical β3α3-hairpin clamp, such as SsIGPS I107 and D128, and several other hairpin clamps within the four-fold symmetrical βαβ modules consistently shortened doubling time relative to the WT. Similarly, for the hydrophobic stack between SsIGPS F40, I45 and V78, all mutations at I45, except to stop codons, displayed increased fitness. These long-range effects on fitness revealed an unexpected allostery between the αβ-loops at one end of the TIM barrel and the active site at the opposite end. The biological relevance of this allostery is supported by known cross-activating communication between HisF synthase TIM barrel and HisH glutaminase in the histidine biosynthetic pathway[44]. Activities of these two proteins are coupled through a physical interaction between the two enzymes, where the αβ-loops of the HisF TIM barrel dock onto the oxyanion hole of the HisH glutaminase active site[44]. Signal transduction between the two active sites spans the entire length of the TIM barrel and is mediated by correlated motions of several networks of residues within the β-barrel and α-helices[45].

Although the IGPS fitness landscape observed in our experiments in yeast may differ from that experienced during evolution of these thermophilic proteins, we believe that the basic biophysical and structural constraints we observed hold true in the natural environment. Moreover, thermophilic orthologues functioning at mesophilic temperatures provide targets of opportunity to detect and map pathways of allostery leading to beneficial fitness under stress conditions. Bernhardt speculated long ago that consecutive enzymes in metabolic pathways might

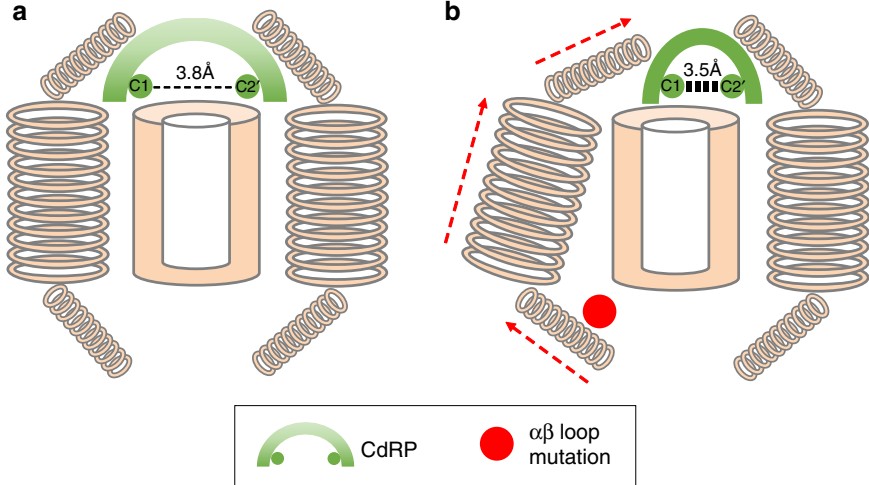

**Figure 8 | The active site of IGPS coordinates the ring closure event converting substrate CdRP to product IGP.** The active site orients the C1 and C2′ atoms of the substrate to a specific distance and geometry, favoring conversion of CdRP to IGP. Correlated motions of grouped amino acids dispersed throughout the protein influence conformation of the active site. This access to the productive enzyme conformation dictates enzyme activity. Under mesophilic conditions, thermophilic IGPS enzymes are less dynamic, less flexible and less active than at their native thermophilic temperatures. (**a**) During yeast growth at 30 °C, the active site of thermophilic IGPS favours an unproductive conformation, where CdRP is oriented in an extended, non-reactive conformation. Reduced activity results in reduced fitness of the WT thermophilic IGPS compared to the WT mesophilic IGPS. (**b**) An allosteric beneficial effect on fitness was observed with certain mutations of the αβ-loop, resulting in greater fitness of the mutants compared to the thermophilic WT IGPS proteins at mesophilic temperatures. These mutations transmit their allostery via correlated protein breathing motions to favour the productive conformation of the active site, where CdRP is oriented optimally in a reactive conformation. Improved fitness observed for β-strand residues whose side chain point to the β-strand/α-helices interface implicates the α-helices (Fig. 3) as the conduit for the allostery between the active site βα-loops and the αβ-loop.

preferentially associate with each other to enhance the through-put of substrates to products[46]. While the assembly status of the three IGPS in our study is monomeric, various combinations of bifunctional and multifunctional enzymes from the tryptophan biosynthetic pathway occur in nature[47]. Thus, the inherent ability of a TIM barrel to stimulate or enhance the activity of another protein in an enzymatic pathway similar to the HisF/HisH pair through allosteric interactions would be advantageous.

Our fitness data for IGPS suggest that mutations in the αβ-loops induce changes in the distal active site βα-loops to favour the catalytically active conformation (Fig. 8). They may do so by mimicking the physiological stimulation by partner proteins or by inducing the conformation naturally preferred at their respective optimal growth temperatures. The greater fitness observed for mutations in the outward facing side chain positions in the β-barrel (Fig. 3) compared to mutations with inward facing side chains, implicates perturbations in the α-helical shell as the conduit for the allostery. This conjecture is supported by the SCA results that point to a possible pathway of signal transduction between the active site and the αβ-loops in SsIGPS, mediated by intervening helical elements (Supplementary Fig. 8).

Experimental mapping of fitness landscapes has progressed from sampling of some or all combinations of diallelic loci[48,49] towards large-scale mutational scans[9–14,20,36]. Although there is evidence about the roughness of fitness landscapes for small deviations from the WT sequence[6,50,51], our finding of correlations between the fitness landscapes of divergent IGPS orthologues suggests that the common fold and function act together to smoothen the landscape. In fact, low throughput experiments have shown that the effects of mutations on biophysical properties such as Tm and $\Delta\Delta G°$ are largely conserved across homologues of influenza nucleoprotein[52], and modern versus ancestral thioredoxins and β-lactamases[53]. These studies suggest that amino-acid preferences at a given position in the structure are mostly conserved during evolution, contrary to a modelling prediction of Pollock et al.[54] However, these works focused on specific biophysical properties of the proteins, rather their full

functional consequences. This limitation was lifted in a deep mutational scan of two influenza virus nucleoproteins (94% sequence identity) in their full physiological context[14]. Again, the amino-acid preferences were found to be conserved at most sites, suggesting strongly correlated fitness landscapes of aligned positions. In agreement, we found significantly higher correlation of fitness landscapes of aligned positions even when the WT amino acids differed between orthologues pairs (Fig. 4c,d).

Beyond constraints imposed by the structure, ancestral sequence reconstruction of ribonuclease H1 suggests that various molecular mechanisms are employed to stabilize proteins during evolution in response to environmental stress[55]. Diverse distributions of ionic, H-bond, and hydrophobic interactions are observed in our three orthologues[56–58], indicating that each protein sampled different sequence space and evolutionary paths as part of its 'thermodynamic system drift'[55]. At the same time, high correlation of fitness landscapes between positions with identical WT amino acids irrespective of structural alignment (Fig. 4a) highlight the biochemical and physical constraints required for properly maintaining the protein fold.

In summary, we found a very strong correlation of organism-level fitness landscapes of three extant proteins at 30–40% sequence identity, and of bacterial versus archaeal origin. Two related interpretations of this result appear possible. First, conservation of amino-acid preferences does persist across the two phylogenic domains and low sequence identity. Second, in a complementary way, we interpret this conservation as transloca-tion of fitness landscapes in sequence space: fitness landscapes of single point-mutants can be successfully translocated to a different starting point in sequence space. We propose to visualize translocation by presenting the fitness landscapes of single point-mutants as pinwheels in sequence space (Fig. 9). All mutants are one change away from the WT, with the colour and height of each mutant representing its fitness. In this analogy, translocation means that pinwheels for orthologues centered far apart in sequence space still maintain a similar shape and colour profile.

**a**

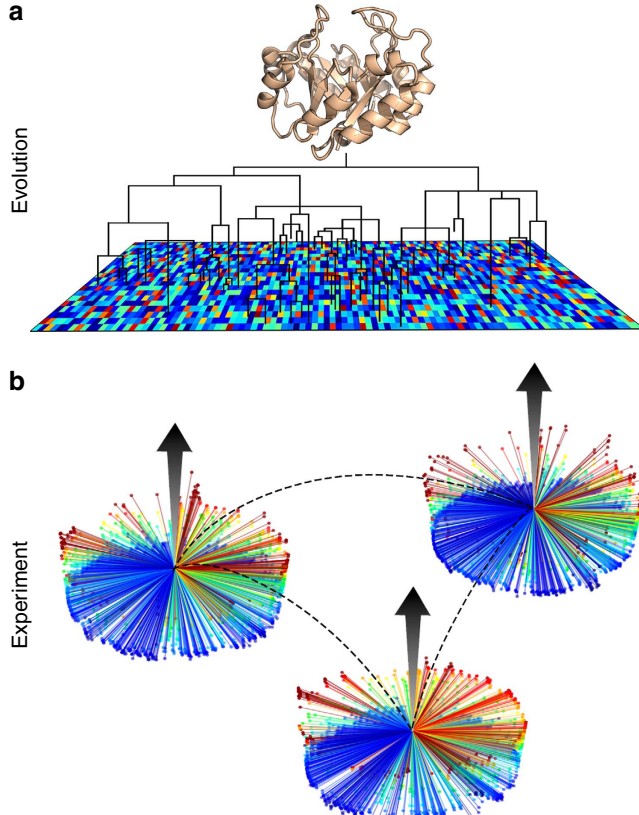

Evolution

**b**

Experiment

**Figure 9 | Translocation of fitness landscapes in the sequence space of orthologous TIM barrels. (a)** Over evolutionary time, orthologous proteins adapt to changing environment for optimal fitness. Steps in sequence space select for protein stability and activity under their native conditions, all while retaining the TIM barrel fold and IGPS function. **(b)** Experimentally derived fitness landscapes mapped from point mutations represent single steps from WT sequence. Despite significant divergence of WT in sequence space, the fitness landscapes of IGPS orthologues remain correlated (dashed lines). Rather than traditional two-dimensional heatmaps, fitness values are displayed on a three-dimensional pinwheel, highlighting the wide range of possible fitness effects of a single sequence step. The profiles of the pinwheels are similar, indicating the correlation of fitness landscapes, even if WT sequences (centers of the wheels) are only ∼40% identical and widely separated. PCA demonstrates a correlation between experimental fitness landscapes and amino-acid preferences in evolved sequences.

Understandably, deviations from the perfect correlation between the landscapes of orthologues result from epistatic effects within the protein. As mentioned previously, the magnitude of epistasis is inversely related with the correlation between the fitness landscapes. For proteins of relatively small divergence, biophysical models achieve a significant power to predict the effects of mutations[59,60], especially if informed by high-throughput mutational scan data[10,15]. Our findings suggest that such approaches may be extended to proteins with a greater degree of sequence divergence. Further development of the models and metrics for comparing the fitness landscapes[61,62] will produce appropriate tools for quantifying the landscapes at various degrees of divergence between sequences.

Translocation of fitness landscape is facilitated by plasticity of the TIM barrel fold, both in natural sequences and *in vitro*, as stable TIM barrels can be created by fusions of natural half barrels, $(\beta\alpha)_4$ (refs 63,64). Indeed, it is well established that the *inverse* protein folding problem, designing sequences for a given template structure, has multiple solutions[65]. Recently, the first

successful *de novo* TIM barrel design employed a completely symmetric four-fold repeat[25,66]. Observations of the four-fold symmetry in both our experimental fitness landscape and *in silico* design serve to validate both approaches in understanding the fundamental properties of TIM protein architecture. From an experimental perspective, fitness studies on other common proteins platforms have the potential to reveal unanticipated sequence-structure-fitness relationships and provide new strategies for *de novo* protein design[25,67].

## Methods

**Strains and culture conditions.** *S. cerevisiae* strain BY4742 Δ*IGPS::KanMX* was produced using the same PCR-generated deletion strategy described by the Saccharomyces Genome Deletion Project[68]. The last 810 bp of the TRP3 gene encoding IGPS were replaced with the KanMX gene. Deletion of IGPS with the KanMX gene was confirmed by Sanger sequencing.

The pRS416 vector carrying the auxotrophic URA3 marker was a gift from Daniel Bolon's lab. A lower expressing Tma19 promoter, also provided by the Bolon laboratory, was used to increase the sensitivity of the fitness assay. Three silent mutations were introduced into the plasmid at the URA3 marker, the ampicillin resistance marker, and the Tma19 promoter to disrupt BSAI recognition sites. The BSAI enzyme was used to create the saturating mutagenesis libraries.

IGPS genes were purchased from Genscript. An N-terminal 6 × His tag and Tev protease recognition site were added to each construct in anticipation of future studies requiring protein abundance measurements and protein purification. In anticipation for *in vitro* folding studies, the non-canonical N-terminal α00 of each gene was deleted to reduce aggregation during refolding of purified proteins (SsIGPS Δ1–26, TmIGPS Δ1–31, TtIGPS Δ1–34). To prevent non-specific cleavage when using Tev protease, position 18 was mutated from arginine to serine in SsIGPS. To prevent disulfide bonds and oxidation by molecular oxygen, position 102 was mutated from a cysteine to a serine in TmIGPS. IGPS genes were cloned into the pRS416 vector using restriction sites, SpeI and BamHI.

For each of the orthologues, we created 8 plasmid libraries where a 10 amino-acid region was mutagenized using the EMPIRIC method, see Hietpas *et al.*[20] for a detailed protocol. Briefly, inverted BsaI restriction sites are introduced by PCR within the region of interest. BsaI digestion is followed by directional sticky-end ligation of oligonucleotide cassettes containing a single randomized site containing all 64 codons. Thus, each library contained 640 sequence variants corresponding to all possible amino-acid (and codon) mutations at each position. The sequence variance was deterministic, as no error-prone polymerases were used.

Yeast transformation was performed using the LiAc/SS carrier DNA/PEG method as described by Gietz and Schiestl[69]. Yeast cells were grown in rich media with G418 to select for IGPS knockout yeast. Transformed cells were selected on synthetic minimal media lacking uracil. Selection for IGPS activity was achieved through growth of transformed yeast (one plasmid library per culture) in synthetic drop-out medium lacking tryptophan. All growth experiments were performed at 30 °C. Liquid cultures were maintained in log phase throughout the fitness assay by periodic dilution. G418 selection was maintained throughout the growth. The oligonucleotide sequences for mutant library generation and primer sequences used for creating the plasmid libraries and processing the deep-sequencing samples are listed in Supplementary Data 1.

**EMPIRIC data processing.** Illumina deep sequencing (36 base single reads) was performed by Elim Biopharmaceuticals. In-house analysis of the deep-sequencing results was performed using custom software. Reads were stringently filtered based on several criteria: Phred score > 20 across all 36 bases (error probability of 0.01), valid barcode match, single reference sequence match for anticipated single codon mutations, and absence of MmeI recognition site. MmeI enzyme was used to create overhangs for ligation of time-stamping barcodes. Sequence counts were tracked on the nucleotide and amino-acid level. A total of 5,040 (10 AA/library × 8 libraries/orthologue × 3 orthologues × 21 mutation types) amino-acid mutations were created. Raw fitness was calculated as the slope of the $\log_2$ relative abundance of the mutant to WT versus time over 6 to 10 time points spread over 4 doubling periods of the yeast,

$$w_i = \frac{\mathrm{d}}{\mathrm{d}t}\log_2\left(\frac{N_i(t)}{N_i^{\mathrm{WT}}(t)}\right),\qquad(1)$$

where $N_i(t)$ is the abundance (count) of mutant $i$ at time $t$, and $N_i^{\mathrm{WT}}(t)$ is the count of WT amino acid at the corresponding position. Slope was determined using the linear regression, Supplementary Fig. 2. From the raw fitness $w$, selection coefficients were determined according to

$$s = -\frac{w}{w_{\mathrm{STOP}}},\qquad(2)$$

where $w_{\mathrm{STOP}}$ is the average raw fitness of all mutations to a stop codon within a 10 amino-acid region, corresponding to an individual selection experiment, $w_{\mathrm{STOP}} = \frac{1}{10}\sum_{i=1}^{10} w_{i,\mathrm{STOP}}$. This normalization ensures that on average, stop codons have a selection coefficient of − 1. Fitness calculations were performed on all values

except 79 mutations leading to MmeI recognition sites. Seven mutations had poor coverage in our mutagenesis libraries and were also removed from analysis: SsIGPS A174W, TmIGPS I131M, TmIGPS I178C, TmIGPS I178W, TtIGPS G78M, TtIGPS L153C and TtIGPS L153Y. After filtering, a total of 4,954 mutations were analysed.

To assess the reproducibility of our fitness results, we compared selection coefficients of two full biological replicates of regions β3, β4 of SsIGPS comprising 399 mutations ($R = 0.947$, Supplementary Fig. 9a).

**Sequence and structural analyses.** Pairwise sequence alignments of the three orthologues were performed using Clustal Omega provided by EMBL-EBI[70]. Sequence identities and similarities were calculated using Ident and Sim Program in the Sequence Manipulation Suite provided by bioinformatics.org. PDB accession codes used for structural analyses were 2C3Z for SsIGPS[56], 1I4N for TmIGPS[57] and 1VC4 for TtIGPS[58]. RMSDs of structural alignments were performed using SPalign[71]. Structural figures were generated using PyMOL Molecular Graphics System v1.8.2.0. Accessible surface area (ASA) was calculated using EBI PISA tool. RSA was calculated from ASA using an empirical scale[24].

**Comparison of fitness distributions.** For structural analyses, specific groups of residues were compared to identify differences in their distributions of fitness effects. The fraction of beneficial mutations was used to describe the shape of fitness distribution. A permutation test was used to determine if two fitness distributions differed significantly. For each subset, the original fitness measurements for 19 amino-acid substitutions at a given position were reassigned to a random position and the corresponding fraction of beneficial mutations was determined. Statistical significance ($P$ value) of the observed fraction of beneficial mutations was calculated from a distribution ($N = 10,000$) of the randomized values. Plots were generated in R version 3.2.0.

**Correlations of fitness landscapes.** The Pearson correlation coefficient was used to assess the similarity in response to the set of 19 amino-acid substitutions for a given pair of residues. To assess the statistical significance, we compared the fitness landscapes of two full biological replicates of regions β3, β4 of SsIGPS comprising 20 positions (Supplementary Fig. 9b). The average correlation between fitness landscapes of the replicates was $R = 0.89$, much higher than all the correlations between the orthologues (Supplementary Fig. 10). The difference between the distributions of correlation coefficients of orthologues and that of the replicates was highly statistically significant (KS test, $P < 10^{-4}$).

**Principal component analysis.** PCA was used to identify major sources of variance in our data set. PCA analyses were performed using custom scripts written in Python version 2.7.6. For the EMPIRIC PCA, fitness values were normalized for every position to have the same mean and s.d. PCA was performed separately for each of the orthologues. For the IGPS PCA, the multiple sequence alignment (MSA), 1,744 representative sequences was obtained from PFam (accession code PF00218). Two positions in the fifth library of SsIGPS had no structural match with positions of the TmIGPS and TtIGPS libraries. Therefore, three separate alignments were created using positions corresponding to all 80 library positions in each of the orthologues. The minimal pairwise overlap between the three alignments is >60 positions. Amino-acid frequencies corresponding to our library positions were extracted and the resulting three matrices of size 80 × 20 were log-transformed and normalized for PCA application. No significant differences in the IGPS-PCA results were observed between the three sequence alignments. For the TIM PCA, structural alignment of 71 non-redundant TIM-barrel proteins[72] was constructed using SPalign[71]. Pairwise structural alignment was performed for each of template structures SsIGPS, TmIGPS, and TtIGPS to the 71 representative TIM barrel structures. As with the IGPS-PCA, we extracted the corresponding 80 library positions from the pairwise alignment to obtain three multiple structure sequence alignments (MSSA) for each of the templates. No significant differences in the TIM-PCA results were observed between the three structural alignments.

**Statistical coupling analysis.** SCA was used to identify groups of co-evolving residues that have functional roles in protein activity and stability. A modified IGPS-MSA was generated for the SCA in order to minimize large gaps in the alignment. The full-length sequence of SsIGPS was used as a BLAST search seed in the NCBI non-redundant database. COBALT[73] was used to align 1,000 protein sequences with length between 200 and 300, and redundancy filter reduced number of sequence down to 537 (35 to 95% sequence identity). Aligned positions with >80% gaps were removed from the alignment. This modified alignment was passed into SCA v5.0 MATLAB toolbox[32] and analysed using the default parameters. Two eigenvalues from the SCA positional correlation matrix exceeded the significance threshold. Therefore, the expected number of sectors was set to $k_{max} = 2$. Spectral analysis of SCA sequence correlation matrix revealed no phylogenetic and/or sequence sampling biases, permitting functional interpretation of the two identified protein sectors.

**Data availability.** All data, primer sequences, alignments, and scripts for data analyses are available at https://github.com/yvehchan/TIM_EMPIRIC or from the corresponding author upon request.

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

## Acknowledgements

This research was supported by NIH GM23303 to C.R.M. We thank Daniel Bolon for providing valued expertise, material, and comments on the manuscript. We thank current/past members of the Bolon lab: Ryan Hietpas, Benjamin Roscoe, Parul Mishra and Li Jiang, and Sagar Kathuria from the Matthews lab for insightful discussions during our experimental setup. We thank Troy Whitfield for insightful discussions during our data analysis.

## Author contributions

C.R.M. and Y.H.C. conceived and designed the studies. Y.H.C. performed the experiments. C.R.M. and K.B.Z. guided the data analyses. K.B.Z., Y.H.C. and S.V.V. analysed the data. Y.H.C. wrote the manuscript with contributions from all authors.

## Additional information

**Competing financial interests:** The authors declare no competing financial interests.

