## [Peer Review File · Nature Communications]

Reviewers' comments:

Reviewer #1 (Remarks to the Author):

This study presents a systematic mutagenesis analysis of three enzymes with a TIM-barrel fold, and then analyzes to what extent the fold vs. the sequence determine fitness effects in these enzymes. Overall, I think this is an interesting study, and with some work on the presentation and interpretation side I think I could like it a lot. As it currently stands, however, the work is poorly presented and difficult to figure out.

Specific comments:

1. Most importantly, throughout the manuscript, I found it difficult to figure out what exactly the question is, and what exactly was done. Even figuring out whether this is an experimental or computational study is difficult until one reads the Methods. This may have been aggravated by the fact that I was provided the manuscript without title page, so I couldn't see the title or abstract. Nevertheless, I think the introduction should define a clear question and then outline briefly how this study addresses the question.

2. Related to the previous point, the authors assume that the reader knows the EMPIRIC approach (e.g., l. 30). This is a bad assumption, in particular when submitting to a journal with broad readership, such as Nature Communications. You need to explain clearly what this approach is and what question it answers. I also think you should describe how this approach differs from other deep-mutational scanning approaches that have been recently developed.

3. The deep-mutational scanning literature has exploded over the last 2-3 years, and I feel the manuscript doesn't fully do these developments justice. For example, I saw only one reference by Jesse Bloom (ref 23), and not the most appropriate one, even though he is arguably one of the leading researchers in the deep-mutational scanning field at this time. Please take a look at his recent works in this area.

4. The fitness measure is strange. Normally, neutral mutations are given a fitness of 1, and lethal mutations a fitness of 0. Alternatively, one could work in log-space, in which case 0 would be neutral and -infinity would be lethal. From the short description of fitness calculations in the Methods, and in particular the equation $s = \dots$, I'm wondering whether the authors are actually working with selection coefficients (i.e., relative fitness *differences*) rather than absolute fitnesses. For a selection coefficient, 0=neutral and -1=lethal would make sense.

In either way, this needs to be clarified. The authors also need to explain *how* exactly they normalize fitnesses (lines 417-420). What transformation exactly was used? Please provide the equation. Also, if -1=lethal, what's the meaning of fitness values <-1? And why limit the highest possible fitness to 0.5?

5. In several places in the manuscript, the authors talk about ASA values, but they never describe how these are measured. Also, I tend to think that Relative Solvent Accessibility (RSA) values are generally preferred over ASA values, because the maximum possible ASA values vary widely by amino acid and hence are severely confounded by amino-acid identity (<http://journals.plos.org/plosone/article?id=10.1371/journal.pone.0080635>).

6. lines 150-160: R_mode and R_mean are not defined

7. line 169: "emphasizing that sequence is not the sole determinant of fitness." Why would anybody think this? Experimental work has shown that homologous structures have similar fitness effects (<http://mbe.oxfordjournals.org/content/32/11/2944>), and a comparison of evolutionary divergence has found similar correlation coefficients to the one you found (<https://www.ncbi.nlm.nih.gov/pmc/articles/PMC3678335/>).

8. lines 42-43: "Surprisingly, we found that fitness can be enhanced by mutations distal from the active site in all three orthologs." This observation is consistent with the recent finding that long-range effects of mutations in enzymes are common and affect ~80% of the typical enzyme structure (<https://www.ncbi.nlm.nih.gov/pmc/articles/PMC4854464/>). I would also like to point out that this finding somewhat weakens the point about protein "polarity" made at the beginning of the manuscript. If the TIM barrel were indeed a highly polarized structure, then maybe one would not have made this observation.

9. I did not fully understand the PCA. I think it's not sufficiently clearly described. A biplot would help a lot to figure out what was done.

Signed,
Claus Wilke

Reviewer #2 (Remarks to the Author):

This paper examines the fitness effects of mutations to a small region of three homologs of TIM barrel proteins. The authors use a comprehensive deep sequencing approach to estimate the effects of all of these mutations.

The most significant finding is that the effects of the mutations are substantially correlated even among the distant homologs studied here. This is a significant and important finding, and should be of interest to both protein chemists and evolutionary biologists. I therefore strongly support publication of this paper if the authors can remedy to the two points below.

I have two major critiques that need to be addressed before I support publication:

1) The major finding of this paper is that the effects of the mutations are similar among distant homologs of the TIM barrel protein. However, the authors completely fail to cite the existing literature on the topic of how much mutational effects are similar among homologs, despite the fact that there has recently been a vibrant discussion on this topic. Below I list four references that should not only be cited, but incorporated into the interpretation of the findings. None of these references undermine the novelty of the current paper -- they use different methodologies or look at proteins with very different levels of divergence -- but they are certainly highly relevant:

- a) Pollock et al (PMID 22547823) used simulations to argue that the effects of mutations shift dramatically during evolution, suggesting that there should be little conservation of mutational effects among distant homologs.
- b) Ashenberg et al (PMID 24324165) used low-throughput experiments to dispute Pollock et al, and argue that in fact the effects of mutations are very similar in homologs of the same protein.
- c) Risso et al (PMID 25392342) used low-throughput experiments to also dispute Pollock et al, and argue that in fact the effects of mutations are similar in homologs of the same protein.
- d) Doud et al (PMID 26226986) used deep mutational scanning to examine the effects of all mutations to protein homologs, and argue that the mutational effects tend to be similar. This study appears to be the closest to the current one, except differs in that it examines very close homologs where the current study examines very different ones.

2) There are no replicates for any of the experiments. We are therefore unable to determine how much noise is in the measurements. For instance, the authors compare the correlations between measurements for different homologs -- but we don't know what sort of correlation they would get if they performed full biological replicates of their full experimental process (mutagenesis, transformation, selection, sequencing) on the same homolog. Most recent papers using experimental approaches similar to this one have performed full biological replicates (see for instance: Fig S1 of PMID 25723163; Fig 2B of PMID 27271655; Fig S6 of PMID 25559584). The

use of independent replicates is a basic principle of scientific rigor and experimental design, and the lack of such replicates here is a major shortcoming.

Reviewer #3 (Remarks to the Author):

The authors set out to measure and compare the local fitness landscapes around three distantly related, thermophilic, indole-3-glycerolphosphate synthase enzymes. These proteins share the common TIM barrel protein fold. The introduced all possible individual point mutations at ~80 sites in three orthologs. In all backgrounds, they found mutations of large effect across the protein-- both in the active site and at distal sites. They asked whether the mutations had the same effects in these different genetic backgrounds. They conclude that there is a high degree of correlation between the datasets, and that the local fitness landscapes for these highly diverged homologs are have been conserved over deep evolutionary time.

The dataset is rich and the analysis generally well executed. Their observation that mutations in distant regions of the protein can modulate activity is both fascinating and potentially useful for protein engineers using TIM barrels. Their observation that the effects of mutations correlate across genetic backgrounds is also intriguing and feeds into existing literature on protein evolution. Although--in my view--there are some weaknesses in this part of the analysis, the core observation that the effects of mutations are maintained over long evolutionary time is likely valid. Further, because they place these observations in a structural context, they can begin to understand these patterns in terms of various biophysical constraints on mutations.

Specific Comments:

1. The description of how correlation between landscapes was measured is unclear (starting at line 146, then methods, starting on line 442.). While I was able to determine that it is a Pearson correlation, which mutations go into which comparison was difficult to extract from the text. The language should be clarified.

2. If I understand what the authors did, the histograms in Fig 4 might be slightly deceptive. I think(?) the authors measured the correlation between the effects of all 19 possible mutations at position X in ortholog A with the effects of mutations at position Y in ortholog B. They then repeated this for X/A vs. Z/C and Y/B vs. Z/C, (using position/ortholog notation). Finally, they calculated these three pairwise correlations for all positions within a class of interest (identical wildtype amino acids, for example), and then pooled these correlations to generate the histograms in Figure 4.

If this is true, it seems like a potentially deceptive analysis. They argue that the three landscapes are highly correlated. But, if they pooled their correlation coefficients as above, their signal could reflect correlation between a subset of orthologs rather than all (e.g. mutational effects in orthologs A and B are correlated, but orthologs B and C are poorly correlated). Put another way: do these coefficients reflect conservation of mutational effect across all three orthologs, or tight correlation between a pair of orthologs and weak correlations between the others? Given that the authors argue that structural constraints, rather than specific sequence, determine some of the correlation, this distinction is important. If structural constraints determine correlation, the correlation should exist between all pairwise comparisons, not just in the pooled correlation.

3. The analysis of landscape correlation requires comparing distributions of correlation coefficients. This is both to compare classes of mutations, but also to compare their observed distributions to their null hypothesis. They do so using the mean and the mode of the correlation coefficient distributions. This is insufficient to argue for "significant" correlation between landscapes. The authors should do a formal statistical analysis (such as a Kolmogorov-Smirnov test). Do their observed correlation coefficient distributions differ from a distributions sampled from a null model?

Do the different mutation classes appear to be drawn from the same, or different distributions? This will make their conclusions much stronger.

4. This is a suggestion for consideration. Rather than describing the differences between landscapes solely in terms of correlation, the authors could tie their landscapes back to the classic question of the importance of epistasis in determining fitness landscapes. The authors' analysis reveals to what extent the effects of mutations are independent of background (non-epistatic) or dependent on background (epistatic). Put somewhat imprecisely, if two sites exhibit a correlation of 0.5, 50% of the mutational effects are additive, while 50% are due to epistasis. The authors' data clearly point to extensive epistasis of this sort. Others have observed and discussed "cryptic" epistasis between backgrounds [Lunzer et al. (2010) PLoS Genetics], but the authors' dataset provides a much higher resolution platform to explore the partitioning of variation between additive and epistatic contributions.

5. The authors should more thoroughly discuss the nature of the "fitness" that they measure using the EMPIRIC method. They are observing fitness changes from mutations in thermophilic orthologs under mesophilic conditions in yeast. These orthologs, however, evolved at thermophilic temperatures in a different cellular context. The measured fitness landscape could be quite different from the landscape experienced as the protein evolves. While I understand the technical reasons for measuring "fitness" as the authors did, this mismatch should be explicitly noted and discussed.

Minor comments

6. In the interests of reproducibility, the authors should include the multiple sequence alignment used in their PCA and SCA analyses.

7. I found Fig 9 confusing and (it seems) unnecessary for the argument. In particular, the pinwheel diagrams were insufficiently described to be interpretable.

8. Line 67: "Supplementary Fig 2" should be changed to "Supplementary Fig 3".

9. Line 314: "Simulation" should be changed to "stimulation"

*Reviewers' comments:**Reviewer #1 (Remarks to the Author):*

This study presents a systematic mutagenesis analysis of three enzymes with a TIM-barrel fold, and then analyzes to what extent the fold vs. the sequence determine fitness effects in these enzymes. Overall, I think this is an interesting study, and with some work on the presentation and interpretation side I think I could like it a lot. As it currently stands, however, the work is poorly presented and difficult to figure out.

Specific comments:

1. Most importantly, throughout the manuscript, I found it difficult to figure out what exactly the question is, and what exactly was done. Even figuring out whether this is an experimental or computational study is difficult until one reads the Methods. This may have been aggravated by the fact that I was provided the manuscript without title page, so I couldn't see the title or abstract. Nevertheless, I think the introduction should define a clear question and then outline briefly how this study addresses the question.

The revised introduction highlights our main questions, comparison of fitness landscapes of orthologous enzymes of ancient divergence and low sequence identity, and identifying structural correlates of the TIM barrel fitness landscape (lines 19-21 and 27-28). Lines 29-31 highlight that this is an experimental study.

2. Related to the previous point, the authors assume that the reader knows the EMPIRIC approach (e.g., l. 30). This is a bad assumption, in particular when submitting to a journal with broad readership, such as Nature Communications. You need to explain clearly what this approach is and what question it answers. I also think you should describe how this approach differs from other deep-mutational scanning approaches that have been recently developed.

An outline of the EMPIRIC approach has been added to the Introduction (lines 35-45) and Methods (lines 421-426). In particular, we stress that the EMPIRIC approach does not rely on error-prone DNA replication to introduce sequence diversity. This ensures complete coverage of codon space in the regions of interest, absence of double mutants, while mitigating biological reproducibility issues.

3. The deep-mutational scanning literature has exploded over the last 2-3 years, and I feel the manuscript doesn't fully do these developments justice. For example, I saw only one reference by Jesse Bloom (ref 23), and not the most appropriate one, even though he is arguably one of the leading researchers in the deep-mutational scanning field at this time. Please take a look at his recent works in this area.

We appreciate this comment; references to the recent works by Jesse Bloom and other leaders in the field have been cited and discussed throughout the manuscript and, in particular, lines 336-359.

4. The fitness measure is strange. Normally, neutral mutations are given a fitness of 1, and lethal mutations a fitness of 0. Alternatively, one could work in log-space, in which case 0 would be neutral and -infinity would be lethal. From the short description of fitness calculations in the Methods, and in particular the equation $s = \dots$, I'm wondering whether the authors are actually working with selection

*coefficients (i.e., relative fitness *differences*) rather than absolute fitnesses. For a selection coefficient, 0=neutral and -1=lethal would make sense.*

*In either way, this needs to be clarified. The authors also need to explain *how* exactly they normalize fitnesses (lines 417-420). What transformation exactly was used? Please provide the equation. Also, if -1=lethal, what's the meaning of fitness values <-1? And why limit the highest possible fitness to 0.5?*

In the revised manuscript, fitness is defined as selection coefficient, $s = 0$ for neutral and $s = -1$ for lethal mutants. Terminology has been adjusted as appropriate, to discriminate fitness as a concept and selection coefficient as its quantitative metric. Equations are now provided in Methods, lines 442-452. The linear transformation from the raw slopes of mutant abundance vs time was applied to ensure that the average selection coefficient of all stop codons in a 10-residue library is $s = -1$. The rationale of normalization was that growth competition experiments were performed in batches, with 10 amino acid positions (640 mutant codons) per individual culture. Averaging the stop codon fitness would remove possible differences of overall growth rate between the cultures.

Many cases of “hyperlethal” mutants with $s < -1$ are due to our choice of stop codon fitness normalization as average across the 10-residue region. For example, Ss G126I has $s = -1.727$. However, the average selection coefficient of stop codons at G126 is $s = -1.738$. Therefore amino acids do not appear to deplete faster than stop codons; however, stop codons (*) at G126 depleted much faster than on average (raw slope of log2 abundance of G126* vs time is -0.22, while the average raw slope for all stops within the 10-residue region is about -0.128). Therefore, while deleterious mutants of G126 were not worse than stops at that position, they appear to have $s < -1$ after normalization.

We attempted to recalculate the selection coefficients using the position-specific stop codon decay rate as normalization. While this procedure eliminated a small class of “hyperlethal” mutants such as Ss G126I, the overall noise in the dataset increased, as the stop codon frequencies were no longer averaged across 10 positions. The Pearson correlation coefficient between the complete selection coefficient datasets calculated using the two normalization methods was $R = 0.946$. This is equal to the correlation of selection coefficient of two full biological replicates of regions $\beta 3$, $\beta 4$ of SsIGPS, with $R=0.947$ (new Supplementary Figures 8, 9). Therefore, we decided to keep the analysis as originally performed. Importantly, since normalization is a linear transformation, normalization issues leading to “hyperlethal” mutants due to differences in stop codon decay rates have little or no effect on the reported Pearson correlations of fitness landscapes, or on our overall conclusions.

In the original manuscript, the fitness has been clipped to 0.5 to improve the visual dynamic range of the heatmap in Fig. 2. This limitation has been removed in the revision; analysis has always been performed on the original data without any clipping.

5. In several places in the manuscript, the authors talk about ASA values, but they never describe how these are measured. Also, I tend to think that Relative Solvent Accessibility (RSA) values are generally preferred over ASA values, because the maximum possible ASA values vary widely by amino acid and hence are severely confounded by amino-acid identity
(<http://journals.plos.org/plosone/article?id=10.1371/journal.pone.0080635>).

We repeated the analysis using the RSA metric, which did not change our conclusions. This is reflected in the text, line 97-98. The PISA tool from EMBL-EBI was used to determine ASA, this is now reflected in Methods, line 468.

6. lines 150-160: R_mode and R_mean are not defined

The subsection describing the comparison of fitness landscapes has been rewritten, using the Kolmogorov-Smirnov test and avoiding comparison of mean and median correlations.

7. line 169: "emphasizing that sequence is not the sole determinant of fitness." Why would anybody think this? Experimental work has shown that homologous structures have similar fitness effects (<http://mbe.oxfordjournals.org/content/32/11/2944>), and a comparison of evolutionary divergence has found similar correlation coefficients to the one you found (<https://www.ncbi.nlm.nih.gov/pmc/articles/PMC3678335/>).

We removed the sentence in question and cited the suggested references, line 170.

8. lines 42-43: "Surprisingly, we found that fitness can be enhanced by mutations distal from the active site in all three orthologs." This observation is consistent with the recent finding that long-range effects of mutations in enzymes are common and affect ~80% of the typical enzyme structure (<https://www.ncbi.nlm.nih.gov/pmc/articles/PMC4854464/>). I would also like to point out that this finding somewhat weakens the point about protein "polarity" made at the beginning of the manuscript. If the TIM barrel were indeed a highly polarized structure, then maybe one would not have made this observation.

We generally agree with this comment, and removed the discussion of "polarity". The reference suggested has been cited, line 261.

9. I did not fully understand the PCA. I think it's not sufficiently clearly described. A biplot would help a lot to figure out what was done.

The PCA was used as a complementary method to compare fitness landscapes as well as to compare fitness landscapes with evolved amino acid frequencies, lines 191-193 and 214-217. Biplots are presented in Supplementary Fig. 6.

Reviewer #2 (Remarks to the Author):

This paper examines the fitness effects of mutations to a small region of three homologs of TIM barrel proteins. The authors use a comprehensive deep sequencing approach to estimate the effects of all of these mutations.

The most significant finding is that the effects of the mutations are substantially correlated even among the distant homologs studied here. This is a significant and important finding, and should be of interest to both protein chemists and evolutionary biologists. I therefore strongly support publication of this paper if the authors can remedy to the two points below.

I have two major critiques that need to be addressed before I support publication:

1) The major finding of this paper is that the effects of the mutations are similar among distant homologs of the TIM barrel protein. However, the authors completely fail to cite the existing literature on the topic of how much mutational effects are similar among homologs, despite the fact that there has recently been a vibrant discussion on this topic. Below I list four references that should not only be cited, but incorporated into the interpretation of the findings. None of these references undermine the novelty of the current paper -- they use different methodologies or look at proteins with very different levels of divergence -- but they are certainly highly relevant:

a) Pollock et al (PMID 22547823) used simulations to argue that the effects of mutations shift dramatically during evolution, suggesting that there should be little conservation of mutational effects among distant homologs.

b) Ashenberg et al (PMID 24324165) used low-throughput experiments to dispute Pollock et al, and argue that in fact the effects of mutations are very similar in homologs of the same protein.

c) Risso et al (PMID 25392342) used low-throughput experiments to also dispute Pollock et al, and argue that in fact the effects of mutations are similar in homologs of the same protein.

d) Doud et al (PMID 26226986) used deep mutational scanning to examine the effects of all mutations to protein homologs, and argue that the mutational effects tend to be similar. This study appears to be the closest to the current one, except differs in that it examines very close homologs where the current study examines very different ones.

We appreciate this critique. We incorporated this argument into the discussion, lines 336-348.

2) There are no replicates for any of the experiments. We are therefore unable to determine how much noise is in the measurements. For instance, the authors compare the correlations between measurements for different homologs -- but we don't know what sort of correlation they would get if they performed full biological replicates of their full experimental process (mutagenesis, transformation, selection, sequencing) on the same homolog. Most recent papers using experimental approaches similar to this one have performed full biological replicates (see for instance: Fig S1 of PMID 25723163; Fig 2B of PMID 27271655; Fig S6 of PMID 25559584). The use of independent replicates is a basic principle of scientific rigor and experimental design, and the lack of such replicates here is a major shortcoming.

We agree with the reviewer that the reported correlations between the fitness landscapes of the orthologs must be gauged against the experimental noise. We have performed two full biological replicates (including independent preparations of plasmid libraries, transformation, growth competition, sample processing and sequencing) of regions $\beta 3$, $\beta 4$ of SsIGPS, comprising 20 residue positions. The selection coefficients found in the replicates were very strongly correlated, $R = 0.947$, Supplementary Fig. 9. Pairwise correlation coefficients between fitness landscapes of the replicates exceeded $R > 0.85$, Supplementary Fig. 9, 10, much higher than ortholog-to-ortholog correlations reported. Observed differences between orthologs were highly statistically significant compared to variation in the two biological replicates. This is now described in the text, line 165 and Methods, line 470. The low variation between replicates was similar to the one previously reported for EMPIRIC approach (PMID 23376099) in yeast. In contrast, in viral systems, variation is generally higher (e.g. PMID 27271655 and PMID 26656922).

Reviewer #3 (Remarks to the Author):

The authors set out to measure and compare the local fitness landscapes around three distantly related, thermophilic, indole-3-glycerolphosphate synthase enzymes. These proteins share the common TIM barrel protein fold. They introduced all possible individual point mutations at ~80 sites in three orthologs. In all backgrounds, they found mutations of large effect across the protein--both in the active site and at distal sites. They asked whether the mutations had the same effects in these different genetic backgrounds. They conclude that there is a high degree of correlation between the datasets, and that the local fitness landscapes for these highly diverged homologs are have been conserved over deep evolutionary time.

The dataset is rich and the analysis generally well executed. Their observation that mutations in distant regions of the protein can modulate activity is both fascinating and potentially useful for protein engineers using TIM barrels. Their observation that the effects of mutations correlate across genetic backgrounds is also intriguing and feeds into existing literature on protein evolution. Although--in my view--there are some weaknesses in this part of the analysis, the core observation that the effects of mutations are maintained over long evolutionary time is likely valid. Further, because they place these observations in a structural context, they can begin to understand these patterns in terms of various biophysical constraints on mutations.

Specific Comments:

1. The description of how correlation between landscapes was measured is unclear (starting at line 146, then methods, starting on line 442.). While I was able to determine that it is a Pearson correlation, which mutations go into which comparison was difficult to extract from the text. The language should be clarified.

We have clarified the definitions, lines 146-154: We compared the fitness landscapes of the orthologs by calculating the Pearson correlation coefficient between fitness values of the 20 mutant amino acids at a pair of positions in two orthologs. The probability distribution of correlation coefficient for a specific set of positions was then used to assess the similarity of fitness landscapes. We use the mode of the distribution, R_{mode} , to characterize a typical strength of correlation between the two fitness landscapes. We considered the following four sets of positions: (1) identical WT amino acids in a pair of orthologs irrespective of structural alignment, (2) all structurally aligned positions irrespective of WT amino acid; (3) all structurally aligned position with non-identical WT amino acids, and (4) positions aligned by their four-fold symmetry, irrespective of WT amino acid.

2. If I understand what the authors did, the histograms in Fig 4 might be slightly deceptive. I think(?) the authors measured the correlation between the effects of all 19 possible mutations at position X in ortholog A with the effects of mutations at position Y in ortholog B. They then repeated this for X/A vs. Z/C and Y/B vs. Z/C, (using position/ortholog notation). Finally, they calculated these three pairwise correlations for all positions within a class of interest (identical wildtype amino acids, for example), and then pooled these correlations to generate the histograms in Figure 4.

If this is true, it seems like a potentially deceptive analysis. They argue that the three landscapes are highly correlated. But, if they pooled their correlation coefficients as above, their signal could reflect

correlation between a subset of orthologs rather than all (e.g. mutational effects in orthologs A and B are correlated, but orthologs B and C are poorly correlated). Put another way: do these coefficients reflect conservation of mutational effect across all three orthologs, or tight correlation between a pair of orthologs and weak correlations between the others? Given that the authors argue that structural constraints, rather than specific sequence, determine some of the correlation, this distinction is important. If structural constraints determine correlation, the correlation should exist between all pairwise comparisons, not just in the pooled correlation.

The correlations between all three pairs of orthologs are of similar magnitude and have a similar shape of distribution; no pair stands out. The revised Fig. 4 shows the three histograms of pairwise correlations overlaid. In almost all cases, for a given class of mutations, the three distributions of correlations for specific pairs of orthologs are indistinguishable according to Kolmogorov-Smirnov test (lines 157-159 Supplementary Tables 1-3).

3. The analysis of landscape correlation requires comparing distributions of correlation coefficients. This is both to compare classes of mutations, but also to compare their observed distributions to their null hypothesis. They do so using the mean and the mode of the correlation coefficient distributions. This is insufficient to argue for “significant” correlation between landscapes. The authors should do a formal statistical analysis (such as a Kolmogorov-Smirnov test). Do their observed correlation coefficient distributions differ from a distributions sampled from a null model? Do the different mutation classes appear to be drawn from the same, or different distributions? This will make their conclusions much stronger.

We appreciate this suggestion. The Kolmogorov-Smirnov test indeed shows that the distribution of correlation coefficients between a specific set of mutations in a pair of orthologs is statistically significantly different from the null model (all possible pairwise correlations or correlations between randomized fitness values), and from the correlations between two biological replicates. Different mutation classes also have correlations drawn from different distributions, lines 160-164.

4. This is a suggestion for consideration. Rather than describing the differences between landscapes solely in terms of correlation, the authors could tie their landscapes back to the classic question of the importance of epistasis in determining fitness landscapes. The authors’ analysis reveals to what extent the effects of mutations are independent of background (non-epistatic) or dependent on background (epistatic). Put somewhat imprecisely, if two sites exhibit a correlation of 0.5, 50% of the mutational effects are additive, while 50% are due to epistasis. The authors’ data clearly point to extensive epistasis of this sort. Others have observed and discussed “cryptic” epistasis between backgrounds [Lunzer et al. (2010) PLoS Genetics], but the authors’ dataset provides a much higher resolution platform to explore the partitioning of variation between additive and epistatic contributions.

We added a new subsection, *Correlation of fitness landscapes and epistasis* (line 172 and below). We agree that the correlation of fitness landscapes is inversely related to the strength of epistasis. Following the methodology of Lunzer *et al.*, we analyzed the fitness of mutations that transform one ortholog toward another (i.e. mutants of protein A where the mutant amino acid serves as WT in protein B, for example SsIGPS N44E to structurally aligned TtIGPS E40), Fig. 5B. Although most of such mutants were neutral or beneficial, some were strongly detrimental, suggesting

epistatic interactions. Remarkably, the fraction of detrimental mutations in our experiment was very similar to that reported by Lunzer *et al.*

5. The authors should more thoroughly discuss the nature of the “fitness” that they measure using the EMPIRIC method. They are observing fitness changes from mutations in thermophilic orthologs under mesophilic conditions in yeast. These orthologs, however, evolved at thermophilic temperatures in a different cellular context. The measured fitness landscape could be quite different from the landscape experienced as the protein evolves. While I understand the technical reasons for measuring “fitness” as the authors did, this mismatch should be explicitly noted and discussed.

We agree with the reviewer that the fitness we measured (growth rate of yeast at 30°C) likely differs from the natural one. This is now acknowledged in Discussion, lines 315-317.

Additionally, the population genetics of natural evolution (very large population size and number of generations) is different from short *in vitro* selection experiments. Thus, our experimental observations are likely skewed toward mutants with greater magnitude of selection coefficients.

Minor comments

6. In the interests of reproducibility, the authors should include the multiple sequence alignment used in their PCA and SCA analyses.

The alignments have been added to Github:

https://github.com/yvehchan/TIM_EMPIRIC/tree/master/EMPIRIC_POSTPROCESS/MSA_inputs_all

7. I found Fig 9 confusing and (it seems) unnecessary for the argument. In particular, the pinwheel diagrams were insufficiently described to be interpretable.

The following text has been added to Discussion:

We propose to visualize translocation by presenting the fitness landscapes of single point mutants as pinwheels in sequence space, as all mutants are one change away from the WT, with the color and height of each mutant representing its fitness. In this analogy, translocation means that pinwheels for orthologs centered far apart in sequence space still maintain a similar shape and color pattern.

8. Line 67: “Supplementary Fig 2” should be changed to “Supplementary Fig 3”.

References properly matched to figures.

9. Line 314: “Simulation” should be changed to “stimulation”

Typo corrected.

REVIEWERS' COMMENTS:

Reviewer #1 (Remarks to the Author):

The manuscript is very much improved. I have no further comments.

Claus Wilke

Reviewer #2 (Remarks to the Author):

The authors have adequately addressed my substantive concerns in terms of performing experimental replicates and better discussing literature.

I support publication of the paper.

There are some formatting errors in bibliography, for instance with references 12 and 29. The authors might check all references for formatting.

Reviewer #3 (Remarks to the Author):

The claims of the authors remain (basically) unchanged in this version of the manuscript relative to the initial submission. They present a rich set of data that allows them to measure the correlations between the fitness landscapes of orthologs of the same protein. The current submission is much stronger than the previous submission -- both in its rigor and presentation. The authors fully addressed my previous concerns. It also appears they addressed the concerns raised by the other referees (though I defer to those referees' judgement on this).

I believe the work is now ready for publication in Nature Communications. The results are intriguing and of broad interest. The conclusions are well-justified.

We appreciate the constructive critiques and suggestions provided by the reviewers that has made our manuscript more accessible, relevant, and rigorous. Our responses to the specific comments made by the reviewers from the latest review are found below.

REVIEWERS' COMMENTS:

Reviewer #1 (Remarks to the Author):

The manuscript is very much improved. I have no further comments.

Claus Wilke

Reviewer #2 (Remarks to the Author):

The authors have adequately addressed my substantive concerns in terms of performing experimental replicates and better discussing literature.

I support publication of the paper.

There are some formatting errors in bibliography, for instance with references 12 and 29. The authors might check all references for formatting.

We checked the references for formatting errors and made appropriate corrections. We believe the bibliography is now properly formatted.

Reviewer #3 (Remarks to the Author):

The claims of the authors remain (basically) unchanged in this version of the manuscript relative to the initial submission. They present a rich set of data that allows them to measure the correlations between the fitness landscapes of orthologs of the same protein. The current submission is much stronger than the previous submission -- both in its rigor and presentation. The authors fully addressed my previous concerns. It also appears they addressed the concerns raised by the other referees (though I defer to those referees' judgement on this).

I believe the work is now ready for publication in Nature Communications. The results are intriguing and of broad interest. The conclusions are well-justified.